# Expression and Function of *C1orf132* Long-Noncoding RNA in Breast Cancer Cell Lines and Tissues

**DOI:** 10.3390/ijms22136768

**Published:** 2021-06-23

**Authors:** Afsaneh Malekzadeh Shafaroudi, Ali Sharifi-Zarchi, Saeid Rahmani, Nahid Nafissi, Seyed Javad Mowla, Andrea Lauria, Salvatore Oliviero, Maryam M. Matin

**Affiliations:** 1Department of Biology, Faculty of Science, Ferdowsi University of Mashhad, Mashhad 9177948974, Iran; Afsaneh.malekzadeh@um.ac.ir; 2Department of Computer Engineering, Sharif University of Technology, Tehran 11155-11365, Iran; sharifi@sharif.edu (A.S.-Z.); rahmanisaid73@gmail.com (S.R.); 3Surgical Department, School of Medicine, Iran University of Medical Sciences, Tehran 14496-14535, Iran; nahid.nafissi@gmail.com; 4Department of Molecular Genetics, Faculty of Biological Sciences, Tarbiat Modares University, Tehran 14115-154, Iran; sjmowla@modares.ac.ir; 5Department of Life Sciences and Systems Biology, University of Turin, 10126 Turin, Italy; andrea.lauria@unito.it; 6Italian Institute for Genomic Medicine, 10060 Candiolo, Italy; 7Novel Diagnostics and Therapeutics Research Group, Institute of Biotechnology, Ferdowsi University of Mashhad, Mashhad 9177948974, Iran

**Keywords:** lncRNA, breast cancer, TNBC, EMT, miR-29, *C1orf132*

## Abstract

miR-29b2 and miR-29c play a suppressive role in breast cancer progression. *C1orf132* (also named *MIR29B2CHG*) is the host gene for generating both microRNAs. However, the region also expresses longer transcripts with unknown functions. We employed bioinformatics and experimental approaches to decipher *C1orf132* expression and function in breast cancer tissues. We also used the CRISPR/Cas9 technique to excise a predicted *C1orf132* distal promoter and followed the behavior of the edited cells by real-time PCR, flow cytometry, migration assay, and RNA-seq techniques. We observed that *C1orf132* long transcript is significantly downregulated in triple-negative breast cancer. We also identified a promoter for the longer transcripts of *C1orf132* whose functionality was demonstrated by transfecting MCF7 cells with a *C1orf132* promoter-GFP construct. Knocking-out the promoter by means of CRISPR/Cas9 revealed no alterations in the expression of the neighboring genes *CD46* and *CD34*, while the expression of miR-29c was reduced by half. Furthermore, the promoter knockout elevated the migration ability of the edited cells. RNA sequencing revealed many up- and downregulated genes involved in various cellular pathways, including epithelial to mesenchymal transition and mammary gland development pathways. Altogether, we are reporting here the existence of an additional/distal promoter with an enhancer effect on miR-29 generation and an inhibitory effect on cell migration.

## 1. Introduction

Breast cancer (BC) is the leading cause of cancer-related death in women worldwide, with an estimated 2,100,000 new cases and 627,000 deaths in 2018 [1,2]. In Iran, the age distribution of BC is nearly 10 years lower than that of counterparts in many countries [3]. Despite the increased incidence rate, BC is highly curable if diagnosed and appropriately treated at early stages. Clinically, classification of BC is based on histopathological findings of the patients’ tissue samples obtained through surgery or biopsy. This includes the assessment of the receptor status of tumor cells, namely estrogen receptor (ER), progesterone receptor (PR), and human epidermal growth factor receptor 2 (HER2) [4,5]. One important subtype of BC is triple negative BC (TNBC), in which neither of the aforementioned receptors are expressed. TNBC tumors make up 10–30% of all breast cancers and are reported to be the most aggressive form of BC, with high histological grade and increased mitotic count, central necrosis, and margins of invasion. It is also associated with younger age of incidence and poorer prognosis in comparison to the non-TNBC patterns [6], as well as an association with shorter duration of breastfeeding, higher age at first pregnancy, and lower parity [7,8].

Long non-coding RNAs (lncRNAs) are important but poorly conserved transcribed RNA molecules with at least 200 nt in length which are involved in complex biological as well as pathological processes. Their mechanism of action can be summarized in four different ways: (I) serving as a signaling molecule, (II) acting as a molecular decoy and/or sponge, (III) guiding some changes in *cis* or *trans* gene expression, and (IV) acting as scaffolds [9,10]. Dysregulation of several lncRNAs has already been reported in breast cancer. They exert either oncogenic or tumor-suppressive functions with significant impacts on various biological and pathological processes, especially in the progression of malignant tumors [11].

*C1orf132* is an intergenic lncRNA located on 1q32.2 between the *CD34* and *CD46* protein coding genes. *C1orf132* (also known as *MIR29B2CHG*) is the host gene for miR-29b2 and miR-29c, with a molecular size of more than 20 kb. Its promoter is more likely to be hypermethylated in basal-like breast tumors compared to low-grade ones [12]. Bhardwaj and colleagues used the next-generation RNA sequencing technique on a breast cancer progression cell line model and discovered that the tumor-suppressor miR-29c is highly downregulated during the course of breast cancer cell progression toward TNBC. Moreover, their data revealed that higher miR-29c expression is associated with a longer survival rate and lower distant metastasis [13].

Despite its potential importance, very little is known about the expression and function of *C1orf132* in tumor cells and tissues. Here, for the first time to our knowledge, we explored the expression pattern of *C1orf132* in different types of breast cancer cell lines and tissues. We also identified and analyzed a distal promoter area with an enhancing function on miR-29b2/c expression and an inhibitory effect on G2/M phase of cell cycle and cell migration.

## 2. Results

### 2.1. C1orf132 Locus and Its Transcripts

*C1orf132*, also known as the host gene for miR-29b2 and miR-29c, is located between two protein coding genes, *CD34* and *CD46* (Figure 1). There are several transcripts with various lengths and number of exons of *MIR29B2CHG* located between the two aforementioned coding genes. No ortholog gene for *C1orf132* has been reported in mice, mostly because lncRNAs are not conserved during evolution. For this reason, we searched for potential lncRNAs overlapping with miR-29b2 and miR-29c in mouse genome. Interestingly, we discovered a lncRNA with conserved chromosomal location and transcriptional direction in reference to its human counterpart (Figure 1).

The human mir precursor has 5 or 6 exons, with its promoter located (~20 kb) upstream of the miR29b2c (Figure 2, indicated in blue). In addition, we noticed several spliced variants of the transcript, some with different start sites, located upstream of the known promoter for miR-29 host gene variant. The latter finding suggested that the lncRNA could bear more roles rather than just serving as a precursor to generate miR-29. Analysis of the FANTOM5 data set indicated the existence of at least two additional distal potential transcriptional start sites (p1 and p2) for the transcription of the longer forms of *C1orf132*, with the putative p2 promoter containing a CpG island (Figure 2A–C). ENCODE data exhibited the expression and chromatin modifications (H3K27ac and H3K4me3) of the *C1orf132* locus in MCF7 cells, while the active marks were mostly concentrated on two start sites: One as the promoter of miR-29 precursor and the other one as the potential promoter for the longer transcript (Figure 2D–G). According to the ENCODE ChIP-seq data, the latter promoter could interact with more than 80 different transcription factors (Figure 2H–J).

### 2.2. C1orf132 Was Significantly Downregulated in TNBC Tissue Samples

Expression analysis of different breast cancer subtypes in TCGA revealed a downregulation of the longer transcripts of *C1orf132* in TNBC patients. According to the results, the expression level of *C1orf132* transcripts originating from the p2 promoter was significantly decreased in ER/PR negative and TNBC patients compared to the ER/PR positive as well as non-TNBC ones (Figure 3A). To confirm this experimentally, we performed qRT-PCR and measured the level of *C1orf132* long isoform in tumor samples along with their adjacent non-tumor tissues. The real-time PCR results on 52 paired tissue samples confirmed the same pattern of significant (*p* < 0.05) downregulation of *C1orf132* in TNBC vs. non-TNBC samples (Figure 3B). However, we found no significant correlation between this isoform and other subtypes of breast cancer (*p* > 0.05).

### 2.3. C1orf132 Transcribed from the p2 Promoter Localizes in the Nucleus

To confirm the potential activity of distal promoters, two reporter constructs were generated, namely pEGFP-1-p1 and pEGFP-1-p2 (Figure 4A). The constructs were then used for transient transfection of MCF7 cells, along with pEGFP-C1 vector as a positive control (Figure 4B). The level of transfection efficiency was assessed by the percentage of fluorescent cells. The presence of the GFP signal revealed the activity of the putative p2 promoter, in contrast to the predicted p1 promoter, which failed to show any activity (Figure 4C,D). Motivated by this finding, we decided to do functional analysis on the p2 promoter by deleting this region and investigating its potential effects on various cellular behaviors as well as transcriptome via the RNA-seq technique.

To investigate the subcellular localization of *C1orf132* in MCF12A cells, we quantified the level of *C1orf132* as well as *B2M* and U1 transcripts within cytoplasmic and nuclear cellular fractions obtained by Ambion cell fractionation buffer. qRT-PCR data on nuclear and cytoplasmic fractions demonstrated that *C1orf132* was prevalently present within the nucleus (*p <* 0.05). U1 and *B2M* were used as internal controls for nuclear and cytoplasmic compartments, respectively (Figure 4E).

Considering the activity of the p2 region, we decided to target it with three pairs of guide RNAs (Figure 5A). After transfecting MCF12A cells with designed gRNAs, the best sharp PCR product of 438 bps (caused by a genomic deletion of about 2000 bps) belonging to the guide numbers 1 and 3 (Figure 5B) were selected. Genomic real-time PCR on MCF12A and edited colonies confirmed that the colonies were knocked-down and that the colony number 5 had the lowest expression level (*p* value < 0.05) for the long transcript of *C1orf132* (Figure 5C).

Flow cytometry analysis revealed a reduction in cell distribution in S phases of the cell cycle progression of the edited compared to the unedited MCF12A cells (*p* < 0.05, Figure 6A). Data obtained from three different biological repeats demonstrated that the putative p2 promoter knock-down could delay the cell cycle progression and significantly increase the number of cells in the G2/M phase (*p* < 0.001, Figure 6B).

### 2.4. Promoter Deletion Enhanced the Migration Ability of MCF12A Cells

As seen in Figure 6, the initial scratch width was monitored in wound healing experiments in edited (top) vs. unedited MCF12A cells (bottom). The results obtained at time points 0 and 24 h revealed that the migration ability was significantly incresed in edited cells compared to the unedited cells (*p* < 0.05; Figure 6C,D).

### 2.5. Gene Expression Profiling after C1orf132 Knock-Down

The gene expression profiles obtained by RNA-seq in two biological replicates of edited and wild-type cells demonstrated 1565 genes with differential expression (adjusted *p* < 0.001), including 551 upregulated and 1014 downregulated genes (Appendix A). Moreover, two-dimensional PCA graph of the data for the two replicates of *C1orf132* edited and unedited MCF12A cells showed nicely separated replicates of edited cells from wild-type cells (Appendix A). Accordingly, the same data emerged from the heatmap of differentially expressed genes between edited and unedited MCF12A cells (Appendix A).

As lncRNAs could have a general enhancer role on neighboring genes, we first looked at any potential altered expression of upstream and downstream genes of the edited region. According to the RNA-seq analysis, the two neighboring genes, *CD34* and *CD46*, were not affected in the edited cells, however, the expression of mir29b2c precursor declined in edited cells compared to the wild-type cells (*p* < 0.001; Figure 6E). According to GO results, the differential expression analysis of the RNA-seq data revealed several up- and downregulated genes involved in various biological processes, including the positive regulation of vascular endothelial cell proliferation and regulation of endothelial cell chemotaxis to fibroblast growth factor biological processes. *FGF2* (*p* < 0.01) and *FGFR1* (*p* = 0.19) genes, which are responsible for the abovementioned processes, were upregulated more than four-fold in edited cells (Figure 6F). FGF2 is a tumor cell survival factor that helps cells to escape apoptosis. Moreover, in the edited cells the expression fold change for *CDH2* (*N-Cadherin*) as a mesenchymal marker was elevated more than twelve-fold (*p* < 0.05) compared to the unedited cells. In contrast, the expression of *CDH1* (*E-Cadherin*) as an epithelial marker was noticeably downregulated in edited cells (*p* < 0.001) compared to the wild-type MCF12A cells. Here, *ZEB1* as an EMT transcription factor showed about 13-fold upregulation in edited cells (*p* < 0.01; Figure 6F). *ESR1*, the gene coding for estrogen receptor alpha (ERα), was downregulated (*p* < 0.001), along with gap junction β-2 (GJB2/CX26; *p* = 0.27) and *WNT4* (*p* < 0.001) in edited MCF12A cells to near zero.

## 3. Discussion

lncRNAs are the largest class of transcripts in human cells, and also the least understood ones. Among various mechanisms attributed to them, being the host of small RNAs, including miRNAs, is the best-known function of these regulatory molecules. However, it is not clear if this type of lncRNAs have additional functions other than being simply the host genes for small RNAs. *C1orf132* or *MIR29B2CHG* is the host gene of two important miRNAs, miR-29b2 and miR-29c. Here, we investigated whether this lncRNA could also have other functions in human breast cancer cells.

There are few reports on *C1orf132* gene and its cellular role. *C1orf132* has been reported as one of five candidate genes whose CpG island methylation are being used to predict an individual’s age in forensic medicine [14,15]. Indeed, the *C1orf132* gene provided the highest accuracy in predicting age [16].

In contrast to coding genes, long non-coding genes have low conservation during evolution, which makes it difficult to find their orthologs in other species. Here, we used a simple logic to find the *C1orf132* ortholog gene in mice by looking at genomic location of mice miR-29b2 and miR-29c. As we expected, there was a lncRNA gene in mice hosting both miRNAs. Surprisingly, *C1orf132* and its mouse counterpart have an exact chromosomal position, located between the *CD34* and *CD46* genes, and are transcribed in the same direction in relation to their neighboring genes. Although there is no need for positional genomic conservation of ortholog miRNA host genes, this evolutionary conservation suggests that *C1orf132* might have other functions in addition to being a host gene for miR-29b2c.

Our in-silico studies revealed two potential promoters in addition to the previously known miRNAs precursor promoter [12], as well as two groups of long and short transcripts for *C1orf132*. The reason for generating short and long transcripts of the gene is not known, however, we observed a similar pattern of short and long transcripts for *PSORS1C3* lncRNA in our previous study [17]. There is a possibility that this kind of gene expression regulation by alternative splicing in a tissue- and cell-specific manner is a more general feature of lncRNAs, and their possible dysregulation could contribute to pathogenesis of many cancers [18]. To explore potential functions of the long transcripts of *C1orf132*, we suppressed their expression by excising the distal promoter by CRISPR/Cas9. It has been already claimed that lncRNAs could have an enhancer effect on their neighboring genes [19]. However, we observed no changes in the expression level of *CD34* and *CD46* in edited cells, both in our RNA-seq and real-time PCR data, ruling out an enhancer role of this sequence on its neighboring genes. Interestingly, there exists a significant downregulation of miR-29c-3p in edited cells, suggesting an enhancer role of the distal promoter on the proximal promoter of the gene. The latter finding makes it very difficult to interpret the RNA-seq results of the edited cells. Indeed, it is impossible to discriminate the direct effects of *C1orf132* from an indirect effect mediated by altered expression of miR-29b2 and miR-29c.

In a bioinformatics analysis of 20 published studies, Yan and colleagues reported that the low expression of miR-29a/b/c is associated with poor prognosis of malignant neoplasms and could be used as a key biomarker to predict cancer progression and recurrence [20]. Furthermore, these miRNAs exhibit both tumor suppressive and oncogenic roles in different cancers. This might be due to the differences in sample types, location, size, or the period of follow-up studies [20,21,22].

We also observed an enhanced migration ability in edited cells, suggesting an invasion suppressor role for *C1orf132*. This is in agreement with Zhang et al. who reported a significant increase in invading cells emerging in miR-29-suppressed HepG2 cells [23]. Jiang et al. reported that miR-29c was remarkably decreased in pancreatic cancer cells and that it had an association with shorter overall survival and tumor recurrence in pancreatic cancer patients [24]. It has been well documented that miR-29c is lost as early as the preneoplastic stage of TNBC tumorigenesis, where its downregulation could cause a worse overall survival via regulating several target genes [13].

EMT is characterized by a loss of function of the E-cadherin adhesion protein (CDH1) in epithelial tissues; breast cancer is a good example of this process. In addition to their role in normal cells, classical cadherins (E- and N-cadherins) have a distinguished role in transforming malignant cells as well as tumor progression, in a process termed “cadherin switching” [25,26]. This process is regulated by EMT transcription factors, including the ZEB family. ZEB1 and ZEB2 are EMT transcription factors which synergistically increase tumor invasion and cell migration [27]. ZEB1 helps epigenetic silencing of *CDH1* by bringing several enzymes to the *E-cadherin* promoter for epigenetic induction or by inhibiting the expression of stemness-repressing miRNAs to cause a dynamic transition of non-cancerous stem cells into cancer stem cells (CSCs) and vice versa [28,29]. In this study, we demonstrated that a cadherin switching evidently took place through a downregulation of *CDH1* and an upregulation of *CDH2* by almost 12-fold, as well as by an almost 14-fold overexpression of *ZEB1*.

Interaction of N-cadherin (CDH2) and FGF receptor (FGFR) leads to their stabilization on the cell surface. In this way, N-cadherin increases tumor cell interactions with endothelial and mesenchymal cells [25]. On the other hand, fibroblast growth factor 2 (FGF2), which can induce proliferation of neoplastic cells and hormone independent tumor growth [30], is a tumor cell survival factor that inhibits cell apoptosis through an autocrine secretory loop. In addition, FGF2 can be an activator of angiogenesis during tumor mass growth and metastasis [31]. In this study, we demonstrated that *FGF2* and *FGFR1* were differentially overexpressed in edited MCF12 cells by almost four-fold. By these results, the pathway analysis of our data by Enrichr was nicely validated for upregulated genes.

On the contrary, we evaluated the expression of *ESR1*, *GJB2*, and *WNT4* genes using qRT-PCR and found that these genes are expressed at very low (near zero) levels in edited cells. GJB2 and ESR1, with a role in “mammary gland development pathway-pregnancy and lactation,” according to WikiPathways, were among the significantly downregulated genes in edited MCF12A cells. Estrogen receptor α (encoded by *ESR1*), which is expressed in normal breast epithelium, has an essential growth and differentiation role. It is also associated with the growth and survival of breast epithelial cancer cells [32], as well as playing a role in the initiation and progression of estrogen-dependent breast cancers [33]. As mentioned earlier, a significant progressive loss of miR-29c, as a result of breast cancer cell line progression model to TNBC, was expected [13]. Therefore, a decline in *ESR1* expression is in line with miR-29c downregulation, which we have observed in our edited cells. We also observed a reduction of *GJB2* in our edited cells. *GJB2* encodes a member of the gap junction protein family often found in the epithelial cells of the mammary duct (luminal) and can form a gap connection channel in the form of homodimers or heterodimers [34]. In breast cancer, the protein was shown to be a tumor-suppressor gene and its very low expression in breast tumor cells might be associated with hypermethylation [35].

Our results revealed a downregulation of *WNT4* in *C1orf132* knocked-down MCF12A cells. Wnt-4 is a member of the WNT family, which encodes important signaling proteins and plays a vital role in some developmental and oncological processes. Several reports have shown an association of Wnt-4 to epithelial cells, such as its expression in normal murine keratinocytes [36], and also throughout skin [37] and mammary gland development [38]. Saitoh and colleagues reported that in keratinocyte cell lines, a poor differentiation and more malignant phenotype is related to the loss of *Wnt-4* gene expression [36]. In another study, a similar association between Wnt4 and mesenchymal to epithelial transition during kidney development has been shown [39]. In contrast, Vouyovitch and colleagues reported that depletion of WNT4 in MCF12 cells inhibited cellular proliferation. Moreover, upregulated WNT4 in non-malignant breast cells stimulated growth, inhibited apoptosis, and increased cell migration and EMT [40].

As the only reference on *C1orf132* lncRNA and cancer, Peng et al. detected two lncRNAs (*C1orf132* and *TMPO-AS*) as a prognostic signature for lung adenocarcinoma patients. By pathway analysis, they suggested that a downregulation of *C1orf132* was associated with a poor prognosis in lung cancer patients, probably by deregulating the “cell cycle and cell adhesion molecules” pathways in cancer cells [41]. Finally, our qRT-PCR data on *C1orf132* knocked-down cells were in line with the RNA-seq and bioinformatics data. Altogether, our data suggest a dysregulated cell cycle and cell adhesion in breast cell line, which might be the cause of aberrant migration of edited MCF12 cells.

As mentioned before, the miR29b2/c precursor is transcribed by the proximal promoter of the gene. To differentiate between the function of *C1orf132* and those of miR29b2/c, we excised the distal promoter of the gene responsible for the generation of longer transcripts of *C1orf132*. Nevertheless, our data revealed that the distal promoter, and hence the longer transcripts, have an enhancing effect on the proximal promoter. Therefore, some of our obtained functional analysis data in the edited cells could simply be some unavoidable secondary effects of the distal promoter on miR29b2/c expression alteration. A precise differentiation between the function of c1orf132 and those of miR-29b2 and miR-29c requires more experiments, including miR-29b2 and miR-29c knockout and/or their overexpression in *C1orf132* knockout cells.

## 4. Materials and Methods

### 4.1. Ethical Statement

This research involved collecting human tissues from Khatam-ol-Anbia and Rasule-Akram hospitals with no experimenting on human subjects or animals. In vitro experiments on commercial cell lines and pathological samples were approved by Ferdowsi University of Mashhad (code number: IR.UM.REC.1399.104). The patients’ written informed consents were collected prior to participation.

### 4.2. Clinical Tissue Samples

In this study, a total number of 52 pathological tissue samples with a diagnosis of primary breast cancer were collected. None of the patients had been treated with preoperative radiotherapy, chemotherapy, or other relevant modalities. Breast cancer tissue samples along with their matched adjacent apparently normal tissues were collected, immediately preserved in liquid nitrogen, and then stored at −80 °C until analysis. All patients signed informed consents and agreed to the use of their surgical specimens for research.

### 4.3. Bioinformatics Analysis

We used the UCSC genome browser to scan the genomic area around *C1orf132* (located at chr1: 207,978,592–208,052,441 (hg19)) for potential promoter activity. Histone modifications by ChIP-seq from ENCODE (H3K27ac and H3K4me3) and DNA-seq were used to detect the active area near the transcription start site (TSS). Using ENCODE ChIP-seq data for different transcription factors (TFs), a list of TFs which bind to *C1orf132* potential promoters was extracted. FANTOM5 data were used to determine which genomic regions the reads originated from.

To evaluate the correlation of *C1orf132* transcripts originating from the putative p2/distal promoter with the tumor state of breast tissues, we performed data mining of The Cancer Genome Atlas (TCGA) breast cancer subtypes to study their association with the expression level of *C1orf132*.

### 4.4. Cloning the Putative Promoters of C1orf132

The potential promoter regions (p1 and p2) for *C1orf132*, according to the bioinformatics analysis, were amplified from genomic DNA using Accu *Taq* polymerase kit (Invitrogen, Carlsbad, CA, USA) and specific primers (Appendix A). The amplified products (1203 bp for p1 and 2143 bp for p2), were cloned into the promoterless green fluorescent protein (pEGFP-1) reporter vector using flanking sequences on primers for *Age*I/*Sal*I restriction sites (Appendix A). The accuracy of the cloned constructs was confirmed by DNA sequencing (Eurofins Genomics Service, Milan, Italy).

### 4.5. Cell Culture and Transfection

MCF7 cell line was cultured in high glucose Dulbecco’s modified Eagle’s medium (DMEM) supplemented with 10% fetal bovine serum (FBS) and 1% penicillin/streptomycin. MCF12A cells were cultivated in a special medium containing DMEM/F12 (Thermo Fisher Scientific, Waltham, MA, USA), 5% heat inactivated horse serum (Gibco, Grand Island, NY, USA), 20 ng/mL recombinant human EGF (AF100-15 Peprotech, Rocky Hill, NJ, USA), 0.5 µg/mL hydrocortisone, 100 ng/mL cholera toxin (C-8052, Sigma, St. Louis, MO, USA), 10 µg/mL human recombinant insulin (Zinc solution 12585-014, Gibco, TN, USA), and 1% penicillin/streptomycin and incubated at 37 °C with 5% humified CO_2_. To examine the promoter activity of the cloned regions, the recombinant vectors were used to transfect MCF7 cells. All experiments were done with complete medium in at least triplicates.

### 4.6. Fractionation Assay

We examined the nuclear vs. cytoplasmic subcellular localization of *C1orf132* in MCF12A cells by fractionation with cell fractionation buffer (Ambion, Austin, TX, USA), according to the manufacturer’s instructions. RNA was extracted to assess the relative proportion of *C1orf132* in the nuclear and cytoplasmic fractions. The transcription levels of beta 2-microglobulin (*B2M*) as a cytoplasmic marker, U1 as a nuclear marker, and *C1orf132* were then assessed using quantitative reverse transcription-polymerase chain reaction (qRT-PCR).

### 4.7. Promoter Activity Reporter Assay

MCF7 cells were seeded in 48-well plates (SPL Life Sciences, Pocheon-si, South Korea). After complete adhesion, cells were transfected with 1 µg of vector (pEGFP-1-p1 or pEGFP-1-p2) using Lipofectamine LTX & PLUS reagent (Invitrogen, Carlsbad, CA, USA), according to the manufacturer’s instructions. pEGFP-1 is a promoterless vector containing an eGFP reporter. The presence of the GFP signal was monitored in MCF7 cells transfected with pEGFP-C1 as the positive control and pEGFP-1-p1 or pEGFP-1-p2 vectors 48 h after transfection and under a fluorescent microscope.

### 4.8. Promoter Deletion Using CRISPR/Cas9 System

In order to suppress *C1orf132* expression, we decided to delete its p2 promoter using the CRISPR/Cas9 system. Three pairs of different guide RNAs (gRNAs) were designed (Appendix A) to target the putative sequences of the p2 transcription start site, using http://crispr.mit.edu (accessed on 5 June 2021). The three gRNAs were then cloned into TOPO-TA gRNA vectors. Briefly, guide RNA plasmid backbone (TOPO-TA gRNA) was linearized using the *Bbs*I enzyme (New England Biolabs, Beverly, MA, USA) and digested at 37 °C for 1 h. Annealing took place using sense and anti-sense oligonucleotides in buffer 2 (NEB), before placing the reaction in a thermocycler with the ramp of 0.1 °C/s from 95 °C to 25 °C. To ligate the gRNA within the linearized vector, T4 DNA ligase (Thermo Fisher Scientific, USA) was used and the reaction was incubated at room temperature for 60 min. Then, *E. coli* cells were transformed with gRNA vectors using heat-shock protocol followed by plating the cells on LB/ampicillin (100 μg/mL) plates overnight. Colony-PCR was performed to verify the sequence of the gRNA plasmids using the reverse oligo of guides and forward primer of TOPO vector.

Different pairs of gRNA-containing vectors were co-transfected along with wild type-Cas9 vector (PX458) into MCF12A cells by lipofectamine LTX & PLUS reagent (Invitrogen, USA). Single cells expressing GFP were selected by a cell sorter machine (BD FACSCelesta) 24 h after transfection. Edited colonies were investigated for deletion by DNA extraction using Puregene Core kit A (Qiagen, Valencia, CA, USA), according to the manufacturer’s instructions, followed by performing PCR using flanking primers (Edited-Test) and DNA sequencing (Europhins, Italy).

### 4.9. Wound Healing Assay

Edited and unedited MCF12A cells were seeded into 24-well plates (with 6 repeats) and grew to 90% confluency. The monolayers were scratched using a 200 μL pipette tip and then the floating cells were removed by several washes with phosphate buffered saline (PBS). Subsequently, the cells were incubated at 37 °C for 24 h before being photographed. The migration area of wound healing was border-lined and analyzed using the ImageJ software (NIH, Bethesda, MD, USA).

### 4.10. Cell Cycle Analysis by Flow Cytometry

Triplicates of three different densities (10,000, 20,000, and 30,000) of edited vs. unedited MCF12A cells were seeded into 12-well plates. The Vybrant™ DyeCycle™ Violet Stain kit (Invitrogen, USA), which is capable of entering living cells and staining DNA, was used to examine the cell cycle 24 h later. Briefly, cells were washed in cold PBS and then harvested and resuspended in the complete medium. Next, flow cytometry tubes each containing 1 mL of cell suspension in complete medium at a concentration of 1 × 10^6^ cells/mL were prepared. In total, 1 μL of Vybrant™ DyeCycle™ Violet Stain was added to each tube (final concentration of 5 μM) and mixed well. After 1 h incubation at 37 °C with protection from light, the samples were analyzed without washing or fixing on a flow cytometer (BD FACSCelesta) using a laser beam.

### 4.11. RNA Extraction and qRT-PCR

A small amount of frozen patient tissue samples was lysed for RNA extraction using RNSol (ROJE, Yazd, Iran) or QIAZOL (Qiagen, USA). Total RNA was extracted according to the manufacturer’s instructions. In total, 1 µg of RNA was first treated with DNase I (Thermo Fisher Scientific, USA) in order to eliminate any traces of DNA contamination and then reverse transcribed using PrimeScript first strand cDNA synthesis kit (TaKaRa, Tokyo, Japan). The expression of target genes was evaluated using BioFACT™ 2X real-time PCR master mix (BioFact, Daejeon, South Korea) through the ABI StepOne real-time PCR system. The sequence of primers used for quantifying each target gene can be found in Appendix A. Relative expression of target genes to *B2M* was calculated according to 2^−ΔΔCt^ method. For the cells, the expression of target genes was evaluated using SensiFAST SYBR No-ROX one-step kit (Bioline Cat. No: BIO-72001, Taunton, MA, USA), according to the manufacturer’s instructions. In total, 30 ng of RNA was used in a one-step real-time PCR using the Rotor-Gene instrument (Qiagen, USA).

### 4.12. TaqMan miR Assay for miR-29c-3p Detection

Hsa-miR-29c-3p (ID 000587) and U6 snRNA (ID 001973) TaqMan miR assays were ordered from Applied Biosystems (Foster City, CA, USA). Total RNA extraction followed by qRT-PCR assay was performed to determine the expression level of miR-29c-3p in edited versus unedited MCF12A cells, according to the manufacturer’s protocol. Briefly, total RNA extraction was performed by Qiazol (Qiagen, USA), followed by reverse transcription using SuperScript II Reverse Transcriptase (Thermo Fisher Scientific, USA), before completing qPCR using Taqman^®^ Universal PCR Master Mix II (Thermo Fisher Scientific, USA). The samples were then incubated for 10 min at 95 °C for initial denaturation and subjected to 40 PCR cycles, each consisting of 95 °C for 15 s and 60 °C for 60 s. U6 was used as the internal control of miR-29c-3p. The 2^−ΔΔCt^ method was used to analyze microRNA levels.

### 4.13. Statistical Analysis

All statistical analyses were performed using GraphPad Prism (GraphPad Software, Inc., La Jolla, CA, USA). Student’s *t*-test was employed to investigate the significance of observed differences of gene expression alterations. All tests were done in 3 biological repeats and values are reported as mean ± standard deviation (SD). *p* values less than 0.05 were considered as statistically significant.

## 5. Conclusions

In summary, using bioinformatics and experimental tools, we discovered the following novel findings: (1) a significant downregulation of the *C1orf132* in triple-negative types of breast cancer; (2) a mouse ortholog of *C1orf132* is located at a similar genomic location as its human counterpart; (3) an additional/distal promoter of *C1orf132* with an enhancer effect on miR-29c generation and an inhibitory effect on cell migration; (4) RNA-seq data on distal promoter excised cells revealed many genes with altered expression involved in various cellular pathways, including mammary gland development.

## Figures and Tables

**Figure 1 ijms-22-06768-f001:**
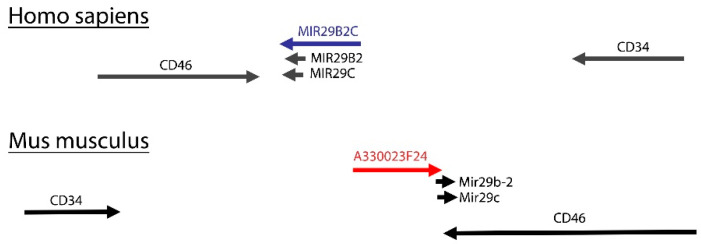
Genomic location of *C1orf132* long-noncoding RNA in humans and mice. Top: The genomic location of miR-29b2, miR-29c, and their precursor, *MIR29B2CHG*, in humans. As shown, the region encoding the lncRNA is located between the *CD34* and *CD46* genes. Bottom: By searching the genomic location of miR-29 in mice, we found out that the lncRNA A330023F24 is the host gene for miR-29b-2 and miR-29c. Interestingly, the chromosomal location and transcription direction of the lncRNA is conserved between humans and mice.

**Figure 2 ijms-22-06768-f002:**
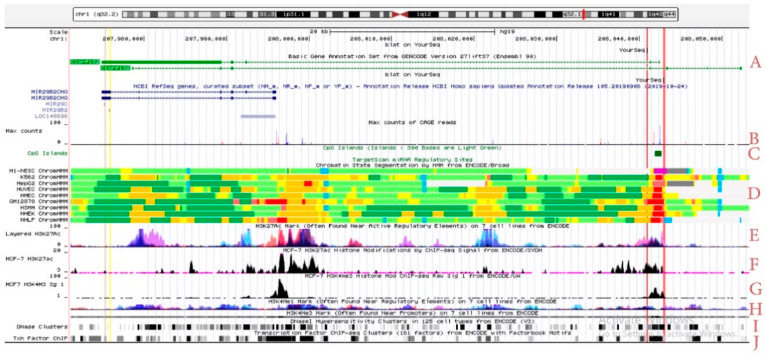
The genomic features of miR29b2c precursor. To make it easier to follow, the location of miR-29b2 and miR-29c are highlighted with yellow and purple color bands, respectively, and the potential distal transcription start sites of *MIR29B2CHG* with red color bands. As it is evident in the figure, there are several spliced variants for the miR29b2c precursor. In addition to the miR29b2c precursor (short variants), there are also longer variants of *C1orf132* (**A**). Other bioinformatics features (UCSC Genome Browser hg19) support the existence of other start sites for *C1orf132* transcription, including: Max counts of CAGE reads belonging to FANTOM5 summary (**B**), CpG island (**C**), active promoter-predicted elements according to Broad ChromHMM (**D**), active or potentially active regulatory elements of layered and MCF7 cells H3K27ac mark (**E**,**F**), active or potentially active promoter and enhancer regions marks by H3K4me3 and H3K4me1, respectively (**G**,**H**), DNase cluster (**I**), and transcription factor ChIP (**J**).

**Figure 3 ijms-22-06768-f003:**
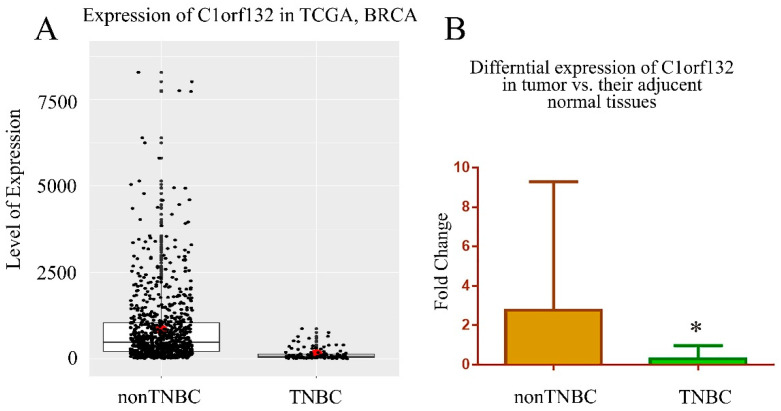
Downregulation of *C1orf132* in TNBC tissue samples. (**A**) The data mining of TCGA was performed for breast cancer samples for the expression of long isoform of *C1orf132* transcribed from the potential distal promoter (ENST00000608023.5_1) on the (–) strand (*p* value < 9.1 × 10^−39^). According to this data, the expression of *C1orf132* was significantly decreased in triple-negative breast cancer (TNBC) samples of BC compared to the non-TNBC samples. (**B**) The qRT-PCR results on 52 fresh tumors and their adjacent apparently normal breast tissue samples demonstrated a significant reduction of *C1orf132* expression in TNBC compared to the non-TNBC tumor tissues (* *p* < 0.05).

**Figure 4 ijms-22-06768-f004:**
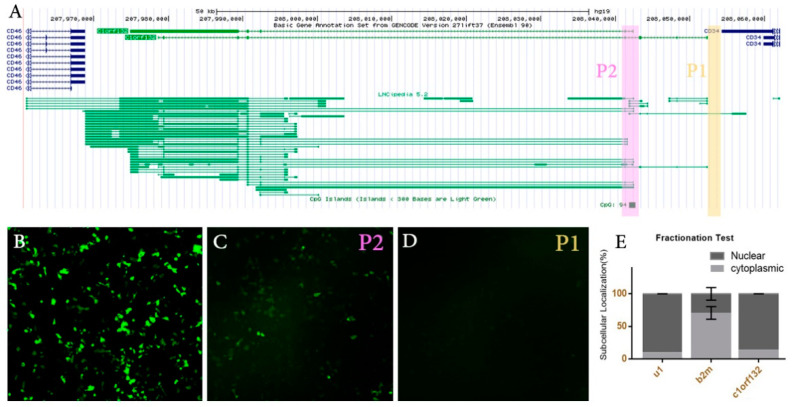
*C1orf132* has at least two active promoters. (**A**) The bioinformatics data suggested the existence of two additional potential promoters (p1 and p2), in addition to its conventional promoter. The predicted sequences correlated to the putative promoters were cloned into a promoterless pEGFP-1 vector. MCF7 cells were transfected with: (**B**) pEGFP-C1 vector as a positive control, (**C**) pEGFP-1-p2 for the putative p2 promoter region, (**D**) pEGFP-1-p1 for the putative p1 promoter region. Note that only the p2 region exhibited promoter activity (100× magnification). (**E**) The subcellular localization of *C1orf132* following nuclear and cytoplasmic subcellular fractionation. *C1orf132* is primarily localized within the nuclear compartment. U1 and *B2M* were used as markers for nuclear and cytoplasmic compartments, respectively.

**Figure 5 ijms-22-06768-f005:**
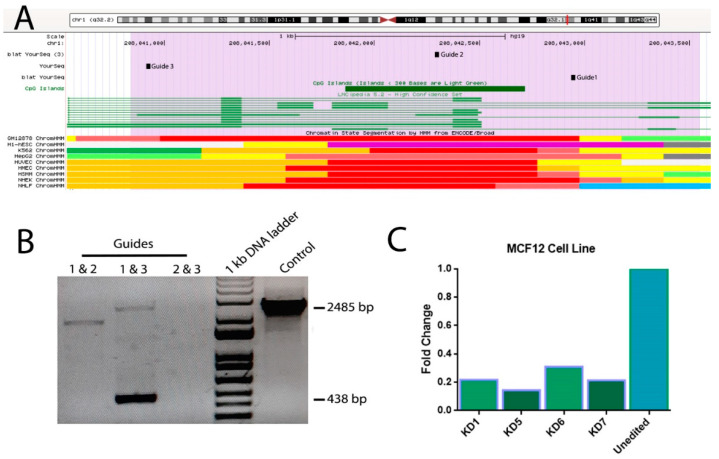
Removing *C1orf132* p2 genomic region using three different guide RNAs. (**A**) The genomic locations targeted with guide RNAs are depicted as guides 1–3 on putative p2 region (highlighted by purple color), (**B**) following transfection of different pairs of guide RNAs into MCF12A cells, using gRNAs 1 and 3 led to the generation of a sharp PCR product of 438 bps caused by a genomic deletion of about 2000 bps. (**C**) Real-time PCR for *C1orf132* long transcript expression on four different MCF12A edited cell colonies in comparison to the unedited cells is shown.

**Figure 6 ijms-22-06768-f006:**
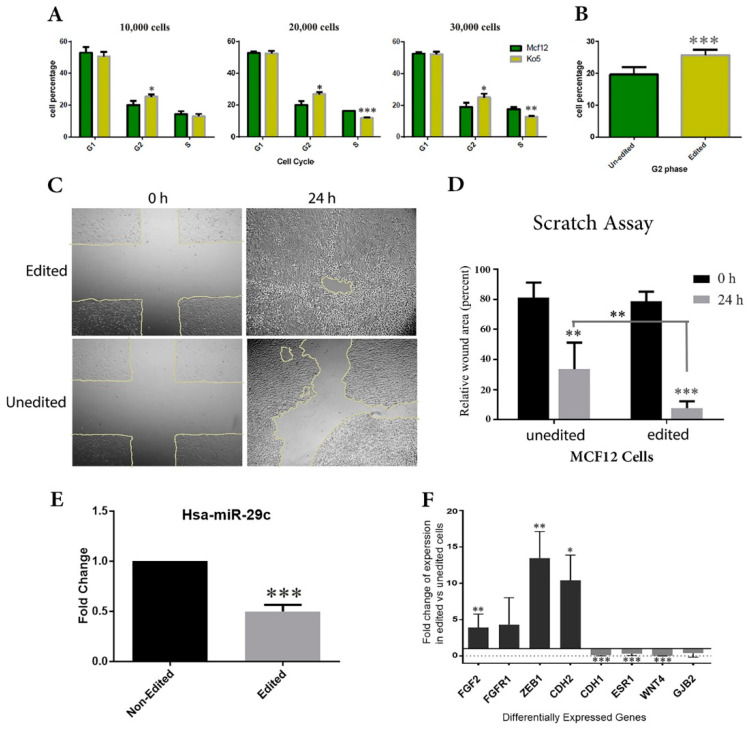
Functional analysis of *C1orf132*. (**A**) Flow cytometry was used to explore cell cycle distribution for edited (*C1orf132*-promoter eliminated) vs. unedited MCF12A cells. (**B**) Data obtained from three different biological repeats demonstrated a G2/M arrest in edited cells (*p* < 0.001). (**C**,**D**) Scratch (wound healing) assay was employed as a tool to monitor cell migration in edited (*C1orf132*-promoter eliminated) vs. unedited MCF12A cells. As it is clearly demonstrated in microscopic pictures, after 24 h the uncovered areas of the plates were significantly decreased (*p* < 0.05) in cells edited for *C1orf132* expression. (**E**) The expression level of miR-29c-3p in *C1orf132*-promoter eliminated cells was significantly reduced by half compared to wild-type MCF12A cells. (**F**) Several genes were differentially expressed in edited vs. unedited MCF12A cells; *FGF2* and its receptor *FGFR1*, *ZEB1* and *CDH2* were overexpressed almost 4 to 15 times in edited cells. Moreover, the expression level of some genes which were mostly epithelial makers, *CDH1*, *ESR1*, *WNT4* and *GJB2* were reduced in edited cells. As expected, the expression level of *C1orf132* decreased as well. Asterisks indicate a significant difference (* *p* < 0.05, ** *p* < 0.01, *** *p* < 0.001).

## Data Availability

The RNA-seq data generated in this study are available at the Gene Expression Omnibus (GEO) database under accession number GSE176304, https://www.ncbi.nlm.nih.gov/geo/query/acc.cgi?acc=GSE176304, accessed on 7 June 2021.

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
