# Peer review of "Expression and Function of C1orf132 Long-Noncoding RNA in Breast Cancer Cell Lines and Tissues"

_ijms, 2021, doi:10.3390/ijms22136768_

Round 1

Reviewer 1 Report

The study by Sharafroudi et al aimed to document the expression of C1orf132, a long non-coding RNA responsible for miR29b/c, in breast tumors and further functional analysis in breast cells. They used bioinformatic analysis combined with experiments in MCF12A cells to parse the functional activity.

In addition they found that the genomic location of C1orf132 location is conserved in human and mice.

This is a well-executed study and the manuscript is well-written. A few comments are noted below.

The primary concern is that there needs to be more clarity regarding the novelty of the finding.

Although the authors mention it, how can their results different between function of C1orf132 and miR29b/c, which we already know is downregulated in cancer. Please expand on this in the discussion?

How does this help us understand the ‘function’ of C1orf132 beyond being the gene that encodes miR29b/c, as the authors suggest? Is it possible to KD miR29b and miR29c and see if there are differences in expression changes noted in Figure 6F, to help differentiate a role for C1orf132 from its miRNAs; or in cells in which p2 is deleted, can expression of those genes be restored (or not) by expressing the miRNAs? Further experiments may assist in delineating the gene from its miRNAs.

Secondary concern is the veracity of the data. The findings would be more compelling if there was a more complete picture of functional outcomes, in addition to differentially expressed genes, perhaps using immunoblotting or confocal imaging. Otherwise this seems to be interesting preliminary data.

Lacking statistical analysis.

Figure 3. For A and B, were the differences significant?

Figure 4. Please clarify the fractionation test. Does this figure indicate that ~5% of U1 expression was in the nucleus and 75% of b2M expression was in the cytoplasm?  It is unclear how this helps understand the localization of C1orf132.

Figure 5C. Were the differences significant?

Figure 6. For A -F, were the differences significant?

Author Response

“The primary concern is that there needs to be more clarity regarding the novelty of the finding.”

We thank the reviewer for the acknowledge of the novelty of the findings. We highlighted the novelty of the work in the last paragraph of the revised introduction and in the concluding paragraph.  

“Although the authors mention it, how can their results different between function of C1orf132 and miR29b/c, which we already know is downregulated in cancer. Please expand on this in the discussion?”

We agree with the reviewer and have now elaborated this point within the discussion of the revised MS. Briefly, miR29b2/c precursor is transcribed by the proximal promoter of the gene. To differentiate between the function of C1orf132 and those of miR29b2/c, we excised the distal promoter of the gene, responsible for the generation of longer transcripts of C1orf132. Nevertheless, our data revealed that the distal promoter, and hence the longer transcripts, have an enhancing effect on proximal promoter. Therefore, there are some unavoidable secondary effects due to the knockout of the distal promoter which mediates by a reduction of miR29b2/c.  

“How does this help us understand the ‘function’ of beyond being the gene that encodes miR29b/c, as the authors suggest? Is it possible to KD miR29b and miR29c and see if there are differences in expression changes noted in Figure 6F, to help differentiate a role for C1orf132 from its miRNAs; or in cells in which p2 is deleted, can expression of those genes be restored (or not) by expressing the miRNAs? Further experiments may assist in delineating the gene from its miRNAs. Secondary concern is the veracity of the data. The findings would be more compelling if there was a more complete picture of functional outcomes, in addition to differentially expressed genes, perhaps using immunoblotting or confocal imaging. Otherwise this seems to be interesting preliminary data.”

As correctly mentioned by the reviewer, there are some ways to differentiate between the function of C1orf132, its short and long variants, its distal and proximal promoters and the expression and function of miR29b2/c. However, and also considering the reviewer’s #2 comment that the MS is already too long we believe these critical suggestions need a variety of experiments and controls and merit a new follow-up research to expand our functional knowledge of C1orf132.  

“Lacking statistical analysis.”

We agree with the reviewer and have now added the statistical analysis within the methods section as well as within the figure legends of the revised MS.

“Figure 3. For A and B, were the differences significant?”

Yes, both differences in figure 3. A & B are significant. We highlighted this in the revised figure.

“Figure 4. Please clarify the fractionation test. Does this figure indicate that ~5% of U1 expression was in the nucleus and 75% of b2M expression was in the cytoplasm?  It is unclear how this helps understand the localization of C1orf132.”

WE thank the reviewer for the question. B2m and U1 were used as internal controls respectively for cytoplasmic and nuclear localizations.  Indeed, they are not entirely detected within one single compartment. Since the processing of miRNAs starts in the nucleus, having some percentages of nuclear detection for a cytoplasmic miRNA is acceptable. Accordingly, about 25% of the cytoplasmic miRNA b2m   was detected in the nucleus compartment. On the other hand, a small percentage (as much as 5%) of nucleic small RNAs, like U1, might be detected in the cytoplasm. As demonstrated in the figure 4E, less than 8% of C1orf132 is detected in the cytoplasm, while the majority of the transcripts were located within the nucleus.

“Figure 5C. Were the differences significant?”

Yes, but we did not show p value for this figure, as they are unique data obtained from our edited cells, but not a representative of a bigger population.

“Figure 6. For A -F, were the differences significant?”

Yes, the differences in figure 6. A-F are significant and we have now added the P-value to the revised figure 6.

Reviewer 2 Report

The issue is interersting but the paper is too long and sometimes not sufficiently clear.

In particular:

The Abstract  is difficult to read and not entirely representative of the manuscript.

The Introduction section does not provide a clear explanation of the rationale . Moreover, lines 48-51 must be deleted.

The Conclusions section does not summarize the principal findings. 

Finally, there are many language mistakes.

Author Response

Reviewer 2

“The issue is interersting but the paper is too long and sometimes not sufficiently clear.”

We thank the reviewer for the acknowledgment that the data present are interesting. We have tried our best to address the reviewer’s comment with the revised MS.

“The Abstract is difficult to read and not entirely representative of the manuscript.”

The abstract is rewritten now, as suggested by the reviewer.

“The Introduction section does not provide a clear explanation of the rationale. Moreover, lines 48-51 must be deleted.”

We agree and revised the MS accordingly.

“The Conclusions section does not summarize the principal findings.”

We have now rewritten the conclusion part, as suggested by the reviewer.

“Finally, there are many language mistakes.”

We have revised the MS and corrected the typo and grammar mistakes within the revised MS.

Round 2

Reviewer 1 Report

Thank you for the revisions. My concerns have been addressed.

Reviewer 2 Report

Authors have modified the text according my suggestions